# $A^*$ Search and Bound-Sensitive Heuristics for Oversubscription Planning

**Michael Katz**[1] , **Emil Keyder**[2]

[1]IBM Research, Yorktown Heights, NY, USA
[2]Invitae Corporation, San Francisco, CA, USA
michael.katz1@ibm.com, emilkeyder@gmail.com

## Abstract

Oversubscription planning (OSP) is the problem of finding plans that maximize the utility value of their end state while staying within a specified cost bound. Recently, it has been shown that OSP problems can be reformulated as classical planning problems with multiple cost functions but no utilities. Here we take advantage of this reformulation to show that OSP problems can be solved optimally using the $A^*$ search algorithm, in contrast to previous approaches that have used variations on branch-and-bound search. This allows many powerful techniques developed for classical planning to be applied to OSP problems. We also introduce novel bound-sensitive heuristics, which are able to reason about the primary cost of a solution while taking into account secondary cost functions and bounds, to provide superior guidance compared to heuristics that do not take these bounds into account. We implement two such bound-sensitive variants of existing classical planning heuristics, and show experimentally that the resulting search is significantly more informed than comparable heuristics that do not consider bounds.

## Introduction

Oversubscription planning (OSP) problems are a family of deterministic planning problems. In contrast to classical planning, where a set of hard goals is specified and the planner searches for a minimal (or low) cost plan that reaches a state in which all of the goals are made true, oversubscription planning specifies a *utility function* that describes the benefit associated with achieving different possible states, and asks for a plan whose cost does not exceed a set bound and achieves as high a utility as possible [Smith, 2004].

While domain-independent classical planning approaches have increasingly standardized around variations on $A^*$ search and heuristics that are automatically extracted from the problem description [Bonet and Geffner, 2001; Keyder and Geffner, 2008; Haslum and Geffner, 2000; Edelkamp, 2001; Helmert *et al.*, 2014; Helmert and Domshlak, 2009], OSP has generally been solved with branch-and-bound algorithms and

heuristics that compute an admissible (in this context non-under) estimate of the utility achievable from a state. In order to obtain these estimates, recent approaches often adapt classical planning techniques such as landmarks [Mirkis and Domshlak, 2014; Muller and Karpas, 2018] or abstractions [Mirkis and Domshlak, 2013], and enhance them with reasoning that is specific to the context of OSP, such as the knowledge that there always exists an optimal plan that ends with a utility-increasing action, or that the cost bound for the problem can be reduced under specific conditions to aid the search algorithm in detecting that improving over the currently achieved utility is impossible.

In contrast to these approaches, our aim here is to show that general methods from classical planning, including $A^*$ search, can be used in the OSP setting nearly as is. This previously turned out to be the case for the related *net-benefit* planning problem, where classical planners solving a compilation were shown to outperform planners designed specifically for that task [Keyder and Geffner, 2009]. Here, we use a similar, recently proposed compilation that converts OSP problems into classical planning problems with multiple cost functions but no utilities [Katz *et al.*, 2019a]. In addition, we demonstrate that existing classical planning heuristics can be used to guide the search for optimal plans. While these heuristics are typically uninformative out-of-the-box, they require only minor modifications (and no specific reasoning about utilities) to render them sensitive to the secondary cost functions and bounds that are introduced by the compilation. Our experiments with $A^*$ and the newly introduced estimators that we refer to as *bound-sensitive heuristics* show that they lead to informed searches that are competitive with, and in some cases outperform, the state of the art for optimal OSP.

One related area of research in the classical setting is that of *bounded-cost planning*, where the planner looks for *any* plan with (primary) cost below a given bound, similar to the treatment of the secondary cost in the OSP setting. Approaches proposed for this setting include dedicated search algorithms [Stern *et al.*, 2011] and heuristics that take into account accumulated cost and plan length at the current search node [Thayer and Ruml, 2011; Haslum, 2013; Dobson and Haslum, 2017]. These approaches work by preferentially expanding nodes in areas of the search space that are likely to have a solution under the cost bound. Optimal OSP, however, requires expanding all nodes that potentially

lie on a path to state with maximal utility. Furthermore, it cannot be assumed that solutions necessarily achieve all soft goals. Heuristics that are able to take into account bounds on secondary cost functions have also been investigated in the stochastic shortest path setting, where they were used as additional constraints in an LP-based heuristic to consider limitations on fuel or time resources [Trevizan *et al.*, 2017].

We now briefly review the various flavors of planning that we consider in this work, and introduce the formalisms by which we describe them.

## Background

We describe planning problems in terms of extensions to the SAS$^+$ formalism [Bäckström and Nebel, 1995]. A *classical planning task* $\Pi = \langle V, O; s_I, G, \mathcal{C} \rangle$ is given by a set of variables $V$, with each variable $v \in V$ having a finite domain $dom(v)$, a set of actions $O$, with each action $o \in O$ described by a pair $\langle \mathsf{pre}(o), \mathsf{eff}(o) \rangle$ of partial assignments to $V$, called the *precondition* and *effect* of $o$, respectively, initial state $s_I$ and goal condition $G$, which are full and partial assignments to $V$, respectively, and the cost function $\mathcal{C} : O \rightarrow \mathbb{R}^{0+}$. A state $s$ is given by a full assignment to $V$. An action is said to be *applicable* in a state $s$ if $\mathsf{pre}(o) \subseteq s$, and $s[\![o]\!]$ denotes the result of applying $o$ in $s$, where the value of each $v \in V$ is given by $\mathsf{eff}(o)[v]$ if defined and $s[v]$ otherwise. An operator sequence $\pi = \langle o_1, \ldots, o_k \rangle$ is applicable in $s$ if there exist states $s_0, \cdots, s_k$ such that (i) $s_0 = s$, and (ii) for each $1 \leq i \leq k$, $o_i$ is applicable in $s_{i-1}$ and $s_i = s_{i-1}[\![o_i]\!]$. We refer to the state $s_k$ by $s[\![\pi]\!]$ and call it the *end state* of $\pi$. An operator sequence $\pi$ is a plan for a classical planning problem if it is applicable in $s_I$ and $G \subseteq s_I[\![\pi]\!]$. The cost of a plan $\pi$ is given by $\mathcal{C}(\pi) = \sum_{o \in \pi} \mathcal{C}(o)$; the goal of optimal classical planning is to find a plan with minimal cost. We refer to a pair of variable $v$ and its value $\vartheta \in dom(v)$ as a *fact* and denote it by $\langle v, \vartheta \rangle$. We sometimes abuse notation and treat partial assignments as sets of facts.

An oversubscription planning (OSP) problem is given by $\Pi_{\text{OSP}} = \langle V, O, s_I, \mathcal{C}, \mathsf{u}, \mathsf{B} \rangle$, where $V$, $O$, $s_I$, and $\mathcal{C}$ are as in classical planning, $\mathsf{u} : (\langle v, \vartheta \rangle) \rightarrow \mathbb{R}^{0+}$ is a non-negative valued utility function over variable assignments (facts), and $\mathsf{B}$ is a cost bound for the plan, imposing the additional requirement that only plans $\pi$ such that $\mathcal{C}(\pi) \leq \mathsf{B}$ are valid. The *utility* of a plan $\pi$ is given by $\sum_{\langle v, \vartheta \rangle \in s_I[\![\pi]\!]} \mathsf{u}(\langle v, \vartheta \rangle)$; the objective of OSP problems is to find valid plans with maximal utility.

A multiple cost function (MCF) problem is given by $\Pi_{\text{MCF}} = \langle V, O, s_I, G, \mathcal{C}_0, \mathscr{C} \rangle$, where $V$, $O$, $s_I$, and $\mathcal{C}_0$ are as in classical planning, $\mathcal{C}_0$ is the *primary cost function*, and $\mathscr{C} = \{\langle \mathcal{C}_i, \mathsf{B}_i \rangle \mid 1 \leq i \leq n\}$ is a set of *secondary cost functions* $\mathcal{C}_i : O \rightarrow \mathbb{R}^{0+}$, and *bounds*, both non-negative. Valid plans for MCF planning problems fulfill the condition $\mathcal{C}_i(\pi) \leq \mathsf{B}_i$ for all secondary cost functions, and optimal plans for MCF planning have minimal primary cost $\mathcal{C}_0(\pi)$. In this paper we only consider MCF problems with a single secondary cost function, i.e. $n = 1$.

## Reformulating OSP Problems

It has recently been shown that an OSP problem can be compiled into an MCF planning problem with a single secondary cost function that corresponds to the cost function $\mathcal{C}$ of the original problem, and is constrained to not exceed the specified bound B [Katz *et al.*, 2019a]. The primary cost function for the problem, or the cost function to be optimized, results from compiling the utilities from the original problem into costs. Two different compilations have been proposed for this task: (i) the *soft goals compilation*, which adds for each variable $v$ that has some value $\vartheta \in dom(v)$ for which a utility is specified, a hard goal, along with actions that are able to achieve this hard goal at different costs, and (ii) the *state delta compilation* which encodes in the cost of each action the change in state utility that results from applying it. Here we consider only (i), as (ii) introduces negative action costs that $A^*$ and existing classical planning heuristics are not designed to handle. Note, however, that our methods do not depend on the specific choice of compilation, as long as they remove utilities from the problem and do not introduce negative action costs.

The *soft goals compilation* was originally introduced in the context of net-benefit planning, which is similar to oversubscription planning but does not specify a bound on plan cost, having instead as an objective the minimization of the difference between the achieved utility and the cost of the plan [Keyder and Geffner, 2009]. It can be applied in the OSP setting to result in an MCF planning problem as follows:

**Definition 1** *Let* $\Pi_{\text{OSP}} = \langle V, O, s_I, \mathcal{C}, \mathsf{u}, \mathsf{B} \rangle$ *be an oversubscription planning task. The* soft goals reformulation $\Pi_{\text{MCF}}^{sg} = \langle V', O', s_I, G', \mathcal{C}_0, \{\langle \mathcal{C}', \mathsf{B} \rangle\} \rangle$ *of* $\Pi_{\text{OSP}}$ *is an* MCF *planning task, where*

- $V' = \{v' | v \in V\}$, *with*

$$dom(v') = \begin{cases} dom(v) \cup \{g_v\} & \mathsf{u}_{max}(v) > 0 \\ dom(v) & otherwise, \end{cases}$$

- $O' = O \cup \{o^{v,\vartheta} = \langle \{\langle v, \vartheta \rangle\}, \{\langle v, g_v \rangle\} \rangle \mid \vartheta \in dom(v), v \in V, \mathsf{u}_{max}(v) > 0\}$

- $G' = \{\langle v, g_v \rangle | v \in V, \mathsf{u}_{max}(v) > 0\}$,

- $\mathcal{C}_0(o) = \begin{cases} 0 & o \in O \\ \mathsf{u}_{max}(v) - \mathsf{u}(\langle v, \vartheta \rangle) & o = o^{v,\vartheta}, \end{cases}$

- $\mathcal{C}'(o) = \begin{cases} \mathcal{C}(o) & o \in O \\ 0 & otherwise, \end{cases}$

*with* $\mathsf{u}_{max}(v) := \max_{\vartheta \in dom(v)} \mathsf{u}(\langle v, \vartheta \rangle)$ *denoting the maximum utility over the values of the variable $v$.*

In the reformulated problem, only the $o^{v,\vartheta}$ actions for which $\vartheta$ is not the maximum utility value of $v$ have positive primary costs. These actions make explicit that a particular utility will not be achieved, and that the plan has instead chosen to achieve the associated $g_v$ by accepting the associated cost penalty. The *primary cost* of a plan $\pi$ for the reformulated problem is then given by $\sum_{v \in V} \mathsf{u}_{max}(v) - \sum_{f \in s[\![\pi]\!]} \mathsf{u}(f)$.

Note that this compilation assumes that utilities are defined for single facts. The more general case, in which utilities are instead defined for logical formulae $\varphi$, can be handled as in the soft goals compilation by introducing a new variable $v_\varphi$, and two actions that achieve its goal value with cost 0 and precondition $\varphi$, and cost $u(\varphi)$ and precondition $\emptyset$, respectively [Keyder and Geffner, 2009]. Since we consider only single fact utilities here, we do not discuss this case in detail.

While this compilation is sound as stated, two further optimizations can be made to reduce the state space of the resulting compiled problem. First, an arbitrary ordering can be introduced over $V$ to ensure that the $g_v$ values are achieved in a fixed sequence, to avoid searching over different orderings. Second, a new precondition fact that is deleted by the $o^{v,\vartheta}$ actions can be added to the original domain actions to ensure that $o^{v,\vartheta}$ actions happen only at the end of the plan and are not interleaved with the original domain actions. We make use of both of these optimizations here.

## $A^*$ **for MCF Planning Problems**

The $A^*$ algorithm extends blind search techniques such as Dijkstra's algorithm by allowing the incorporation of admissible (non-overestimating) heuristics [Hart *et al.*, 1968]. In each iteration of its main loop, $A^*$ picks a node $n$ to expand with minimal $f(n) = g(n) + h(n)$ value, where $g(n)$ is the cost of the path to $n$, and $h(n)$ is an admissible estimate of the remaining cost to the goal. An optimal solution to the problem is found when a node $n$ with minimal $f(n)$ value is a goal node.

To adapt $A^*$ to the MCF planning setting, we store at each node $n$ a set of accumulated path costs $g_i(n)$ resulting from each of the secondary cost functions $\mathcal{C}_1, \ldots, \mathcal{C}_n$, in addition to the accumulated primary cost $g_0(n)$. When a node is taken from the priority queue and expanded, generated successor nodes for which any $g_i(n) > \mathsf{B}_i$ can be immediately pruned, as all $\mathcal{C}_i$ are assumed to be non-negative, and they cannot constitute valid prefixes for solution paths.

One key optimization used in modern $A^*$ implementations in the classical setting is duplicate detection, which allows states that are rediscovered during search to be discarded, if the new $g$ value exceeds the cost of the path to the state that was previously found, or to be updated with a new parent, if the cost of the new path is less. In the MCF setting, care must be taken to ensure that newly discovered nodes are discarded (or replace existing nodes), only when they are dominated by (or dominate), the existing node in all cost dimensions. While the only necessary property of the open list from a correctness perspective is that it order nodes by increasing primary $f(n)$ value, the choice of a secondary ordering heuristic plays a role here: an ordering that causes a dominating node to be generated first and enables subsequently generated nodes to be immediately discarded as dominated results in superior performance. In our implementation of the algorithm, we therefore use an open list that orders nodes by increasing $g_i(n)$ value when their primary $f(n)$ values are the same.

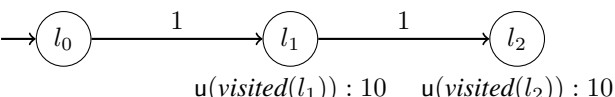

Figure 1: An OSP problem based on the VISIT-ALL domain.

## **Bound-Sensitive Heuristics**

While any admissible heuristic can be used to guide search in MCF planning, classical planning heuristics that ignore bounds entirely are typically extremely uninformative. Consider the problem shown in Figure 1: the agent is initially at $l_0$, and can obtain a utility of 10 by visiting each of the locations $l_1$ and $l_2$. The costs of the actions $move(l_0, l_1)$ and $move(l_1, l_2)$ are both 1. In the compiled MCF version of this problem, an optimal but naive heuristic that ignores the bound will give an estimate for the primary cost of 0, as both $visited(l_1)$ and $visited(l_2)$ can be made true, and the associated 0-primary cost $o^{visited(l_*)}$ actions applied to reach the newly introduced hard goals corresponding to each utility. If, however, $\mathsf{B} = 1$, the optimal $\mathcal{C}_0$ cost at $l_0$ is 10, since $l_2$ cannot be reached at cost $\leq \mathsf{B}$ and the agent must use the $o^{not\text{-}visited(l_2)}$ action to achieve the associated hard goal with a cost of 10. Similarly, if $\mathsf{B} = 0$, the $\mathcal{C}_0$ cost of the optimal plan is 20, since the value of $\mathcal{C}_1$ for all available actions exceeds the bound $\mathsf{B}$. In practice, it turns out that the OSP versions of many classical planning problems have similar behavior: their state spaces are strongly connected, so any variable assignment can be achieved from any state, and classical planning heuristics that ignore bounds are no more informed than blind search.

In order to obtain estimates that take secondary cost bounds into account and can guide heuristic search towards feasible solutions, we therefore introduce *bound-sensitive heuristics*. In the following, we use **b** to denote a *budget vector* of nonnegative reals that indicate the unused component of each of the secondary cost bounds $\mathsf{B}_i$ at a given search node.

**Definition 2 (Optimal bound-sensitive heuristic)** *Given an MCF planning problem* $\Pi_{\mathrm{MCF}} = \langle V, O, s_I, G, \mathcal{C}_0, \mathscr{C} \rangle$*, the* optimal bound-sensitive heuristic $h^*(s, \boldsymbol{b})$ *for a state $s$ and budget vector $\boldsymbol{b}$ is given by the minimal primary cost $\mathcal{C}_0(\pi)$ of a plan $\pi$ for $s$ such that $\mathcal{C}_i(\pi) \leq \boldsymbol{b}_i$ for $i = 1, \ldots, n$.*

By analogy with standard admissible heuristics, an *admissible bound-sensitive heuristic* is a non-overestimating bound-sensitive heuristic:

**Definition 3 (Admissible bound-sensitive heuristic)** *Given an MCF planning problem* $\Pi_{\mathrm{MCF}} = \langle V, O, s_I, G, \mathcal{C}_0, \mathscr{C} \rangle$*, an* admissible bound-sensitive heuristic $h(s, \boldsymbol{b})$ *for a state $s$ and budget vector $\boldsymbol{b}$ is a heuristic $h$ such that $h(s, \boldsymbol{b}) \leq h^*(s, \boldsymbol{b})$ for all $s, \boldsymbol{b}$.*

Any classical planning heuristic that completely ignores $\mathcal{C}_i$ and $\mathsf{B}_i$ can be thought of as an admissible bound-sensitive heuristic that assumes $\mathbf{b} = \infty$. As the value of $\mathbf{b}$ decreases, the value of $h^*(s, \mathbf{b})$ can only increase. In general, it is useful to keep in mind the following property:

**Theorem 1** *Given a state $s$ and budget vectors $\boldsymbol{b}, \boldsymbol{b'}$ such that $\boldsymbol{b} \le \boldsymbol{b'}$ (where $\le$ is interpreted as a pairwise comparison), $h^*(s, \boldsymbol{b}) \ge h^*(s, \boldsymbol{b'})$.*

**Proof sketch:** This follows from the fact that any plan $\pi$ for $s$ such that $\mathcal{C}_i(\pi) \le \mathbf{b}_i$ also has the property that $\mathcal{C}_i(\pi) \le \mathbf{b'}_i$ for $i = 1, \dots, n$ since $\mathbf{b} \le \mathbf{b'}$, yet the opposite is not the case. $\blacksquare$

Theorem 1 applied to MCF planning problems obtained as the *soft goals compilations* of OSP problems states that for any $s$, decreasing $\mathbf{b}$ *increases* $h^*(s, \mathbf{b})$, and decreases the achievable utility, since the primary cost here indicates the utility that the plan must declare unachievable through $o^{v, \vartheta}$ actions with $\mathcal{C}_0(o^{v, \vartheta}) \ge 0$.

## Bound-Sensitive $h^{max}$

The admissible classical heuristic $h^{max}$ estimates the cost of a set of facts $F$ as the cost of the most expensive fact $f \in F$, and applies this approximation recursively to action preconditions in order to obtain the cost of the goal [Bonet and Geffner, 2001]:

$$h_\mathcal{C}^{max}(F, s) = \max_{f \in F} h_\mathcal{C}^{max}(f, s)$$

$$h_\mathcal{C}^{max}(f, s) = \begin{cases} 0 & f \in s \\ \min_{o \in achievers(f, s)} h_\mathcal{C}^{max}(o, s) & otherwise \end{cases}$$

$$h_\mathcal{C}^{max}(o, s) = \mathcal{C}(o) + h_\mathcal{C}^{max}(\mathsf{pre}(o), s)$$

where $h_\mathcal{C}^{max}$ denotes the value of $h^{max}$ computed with a cost function $\mathcal{C}$, and $achievers(f, s)$ denotes the set of actions $o$ for which $f \in \mathsf{eff}(o)$. Note that the $h^{max}$ cost of a fact $f$ that is not present in $s$ is computed by choosing an action $o$ from this set that achieves it with minimum possible cost. Given a set of secondary cost functions and bounds $\mathscr{C} = \{\langle \mathcal{C}_1, \mathsf{B}_1 \rangle, \dots, \langle \mathcal{C}_n, \mathsf{B}_n \rangle\}$, a bound-sensitive version of $h^{max}$ can easily be obtained by replacing the set of achievers used to compute $h_{\mathcal{C}_0}^{max}$ with

$$achievers(f, s)_{\mathcal{C}_0} = \quad \{o \mid f \in \mathsf{eff}(o) \,\wedge$$
$$\bigwedge_{i=1, \dots, n} h_{\mathcal{C}_i}^{max}(o, s) \le \mathsf{B}_i\}$$

where actions $o$ for which any estimate $h_{\mathcal{C}_i}^{max}(o, s)$ exceeds $\mathsf{B}_i$ are not considered. Note that due to the admissibility of $h^{max}$, this restriction of the set of achievers is sound but not complete: it is guaranteed that any action removed from the set of achievers cannot be used in a valid plan, but there may be additional actions that cannot be achievers but are not pruned by the heuristic. In general, any admissible estimate $h_{\mathcal{C}_i}^{max}(o, s)$ could be used to compute $achievers(f, s)_{\mathcal{C}_0}$, but we have chosen $h^{max}$ here for simplicity.

**Theorem 2** *Bound-sensitive $h_{\mathcal{C}_0}^{max}$ is an admissible bound-sensitive heuristic.*

**Proof sketch:** This follows from the admissibility of the heuristic used to compute $achievers(f, s)_{\mathcal{C}_0}$. $\blacksquare$

## Bound-Sensitive Merge-and-shrink

Merge-and-shrink heuristics are a family of abstraction heuristics that incrementally build a representation of the full state space of a problem [Helmert *et al.*, 2014]. The construction process begins with the set of transition systems induced over each state variable; at each step, two transition systems are selected to be merged and replaced with their synchronized product. Since the transition systems need to be represented explicitly in memory, before the merge a shrinking step is perfomed on the two selected transition systems to enforce a user-specified threshold on the size of the synchronized product. This step is performed by abstracting multiple states in the current representation into a single state (and thereby losing optimality). The final output of the algorithm consists of a single abstract transition system in which multiple states and actions from the original task are mapped to a single state or transition, respectively. $h^{\mathrm{MS}}(s)$ is then given by the cost of a shortest path from the abstract state representing $s$ to the closest abstract goal state in the final transition system. This estimate is admissible by definition.

To adapt *merge-and-shrink* to the MCF setting, we maintain for each transition in the abstract state space the minimum $\mathcal{C}_i$ cost for $i = 1, \dots, n$ among all of the transitions from the original task represented by that transition. The distance $\mathcal{C}_i$ between any two abstract states $s, s'$ then represents a non-overestimate of the secondary cost of reaching $s'$ from $s$. A bound-sensitive heuristic value for a state $s$ can be computed as the minimum $\mathcal{C}_0$ cost of a path $\pi$ from $s$ to an abstract goal state $s_g$ whose $\mathcal{C}_i$ cost in the abstract state space does not exceed $\mathsf{B}_i$, for any $i$. The $\mathcal{C}_0$ cost of such such a path can be computed with a modified version of Dijkstra's algorithm that stores secondary cost information for each node and discards nodes for which $\mathcal{C}_i > \mathsf{B}_i$ for any $i$.

**Theorem 3** *Bound-sensitive $h^{\mathrm{MS}}$ is an admissible bound-sensitive heuristic.*

**Proof sketch:** This follows from the fact that the secondary costs used in the abstract state space are the minimums of the secondary costs $\mathcal{C}_i$ of the represented transitions in the original problem, and the proof of admissibility of standard $h^{\mathrm{MS}}$. $\blacksquare$

While the $ms^b$ heuristic can be implemented by running Dijkstra's algorithm in the abstract state space for each heuristic computation, an important optimization when a single secondary cost function is present (which is the case in the compiled OSP problems that we consider) is to run Dijkstra only once during preprocessing, and compute the primary cost in the presence of different bounds on the secondary cost. This information can then be stored as a sequence of pairs $\langle \langle b_0, c_0 \rangle, \dots, \langle b_n, c_n \rangle \rangle$, where $b_0, \dots, b_n$ is strictly increasing and $c_0, \dots, c_n$ is strictly decreasing (recall Theorem 1). $h^{\mathrm{MS}}(s, \mathbf{b})$ is then given by the first $c_i$ such that $\mathbf{b}_i \le \mathbf{b}$.

## Experiments

We implemented our approach in the Fast Downward planner [Helmert, 2006], and evaluated it on a set of publically

| Coverage | 25 | | | | | | 50 | | | | | | 75 | | | | | | 100 | | | | | |
|---|---|---|---|---|---|---|---|---|---|---|---|---|---|---|---|---|---|---|---|---|---|---|---|---|
| | BnB | bl | max$^b$ | max | ms$^b$ | ms | BnB | bl | max$^b$ | max | ms$^b$ | ms | BnB | bl | max$^b$ | max | ms$^b$ | ms | BnB | bl | max$^b$ | max | ms$^b$ | ms |
| airport | **27** | ±0 | -1 | -1 | -9 | -9 | **22** | ±0 | ±0 | -1 | -4 | -4 | **21** | ±0 | -1 | ±0 | -4 | -4 | **21** | ±0 | -3 | -3 | -5 | -5 |
| barman11 | 12 | ±0 | **+1** | ±0 | ±0 | ±0 | **8** | ±0 | ±0 | ±0 | ±0 | ±0 | **4** | ±0 | ±0 | ±0 | ±0 | ±0 | **4** | ±0 | ±0 | ±0 | ±0 | ±0 |
| barman14 | 6 | ±0 | ±0 | ±0 | **+2** | ±0 | **3** | ±0 | ±0 | -3 | ±0 | ±0 | **0** | ±0 | ±0 | ±0 | ±0 | ±0 | **0** | ±0 | ±0 | ±0 | ±0 | ±0 |
| blocks | **35** | ±0 | ±0 | ±0 | ±0 | ±0 | 28 | ±0 | +1 | -2 | **+4** | ±0 | 21 | ±0 | ±0 | ±0 | **+8** | ±0 | 18 | ±0 | ±0 | ±0 | **+8** | ±0 |
| childsnack14 | 0 | ±0 | +1 | ±0 | **+2** | ±0 | **0** | ±0 | ±0 | ±0 | ±0 | ±0 | **0** | ±0 | ±0 | ±0 | ±0 | ±0 | **0** | ±0 | ±0 | ±0 | ±0 | ±0 |
| depot | **16** | ±0 | -1 | -2 | -1 | ±0 | **11** | ±0 | ±0 | -4 | ±0 | -1 | 7 | ±0 | ±0 | -1 | ±0 | ±0 | 4 | ±0 | ±0 | ±0 | **+1** | ±0 |
| driverlog | 15 | ±0 | ±0 | ±0 | ±0 | ±0 | 13 | ±0 | +1 | -1 | +1 | ±0 | 10 | ±0 | +1 | ±0 | **+2** | ±0 | 7 | ±0 | +1 | ±0 | **+4** | ±0 |
| elevators08 | 30 | ±0 | ±0 | -1 | -1 | ±0 | 25 | ±0 | -1 | -1 | ±0 | ±0 | 23 | ±0 | -1 | -1 | **+1** | ±0 | 17 | +1 | ±0 | -1 | **+3** | +1 |
| elevators11 | 20 | ±0 | ±0 | ±0 | ±0 | ±0 | 19 | ±0 | ±0 | ±0 | ±0 | ±0 | 18 | ±0 | -1 | -1 | **+1** | ±0 | 14 | +1 | ±0 | -1 | **+2** | +1 |
| floortile11 | 9 | ±0 | ±0 | ±0 | -2 | ±0 | 4 | ±0 | **+1** | ±0 | ±0 | ±0 | 2 | ±0 | **+2** | **+2** | +1 | ±0 | 2 | ±0 | **+4** | **+4** | ±0 | ±0 |
| floortile14 | 9 | ±0 | ±0 | ±0 | -2 | -3 | **2** | ±0 | ±0 | ±0 | ±0 | ±0 | 0 | ±0 | **+2** | +1 | ±0 | ±0 | 0 | ±0 | **+5** | **+5** | ±0 | ±0 |
| freecell | 77 | ±0 | -14 | -33 | -12 | -6 | 30 | ±0 | -2 | -13 | -2 | -1 | 21 | ±0 | -6 | -6 | -1 | -1 | 20 | ±0 | -6 | -6 | -2 | -4 |
| ged14 | **20** | ±0 | ±0 | ±0 | ±0 | ±0 | **20** | ±0 | ±0 | ±0 | ±0 | ±0 | **20** | ±0 | ±0 | ±0 | -1 | ±0 | **20** | ±0 | ±0 | ±0 | ±0 | ±0 |
| grid | **5** | ±0 | ±0 | -1 | ±0 | ±0 | **3** | ±0 | ±0 | ±0 | -1 | ±0 | **2** | ±0 | ±0 | ±0 | -1 | ±0 | 1 | ±0 | ±0 | ±0 | **+1** | ±0 |
| gripper | 11 | ±0 | ±0 | ±0 | **+1** | ±0 | **8** | ±0 | ±0 | ±0 | ±0 | ±0 | **8** | ±0 | -1 | ±0 | ±0 | ±0 | **8** | ±0 | -1 | ±0 | ±0 | ±0 |
| hiking14 | 19 | ±0 | -1 | -5 | **+1** | ±0 | 14 | ±0 | -1 | -3 | **+3** | ±0 | 13 | ±0 | -2 | -2 | **+2** | ±0 | 11 | ±0 | -2 | -2 | **+3** | ±0 |
| logistics00 | 21 | ±0 | **+1** | ±0 | ±0 | ±0 | **16** | ±0 | ±0 | ±0 | ±0 | ±0 | 12 | ±0 | **+2** | ±0 | **+2** | ±0 | 10 | ±0 | ±0 | ±0 | **+4** | ±0 |
| logistics98 | 6 | ±0 | **+1** | ±0 | ±0 | ±0 | 4 | ±0 | **+1** | ±0 | **+1** | ±0 | 2 | ±0 | **+1** | ±0 | **+1** | ±0 | **2** | ±0 | ±0 | ±0 | ±0 | ±0 |
| miconic | 96 | ±0 | -1 | -4 | **+12** | -1 | 65 | ±0 | ±0 | -1 | **+7** | ±0 | 55 | ±0 | ±0 | ±0 | **+11** | ±0 | 50 | +5 | ±0 | ±0 | **+11** | +4 |
| mprime | **35** | ±0 | ±0 | -2 | -4 | -2 | **28** | -1 | -1 | -5 | -3 | -1 | **24** | ±0 | -1 | -2 | -2 | ±0 | 19 | ±0 | +1 | -5 | -2 | ±0 |
| mystery | **29** | ±0 | ±0 | ±0 | -2 | ±0 | **27** | -1 | ±0 | -3 | -4 | -1 | **21** | ±0 | ±0 | -3 | -1 | ±0 | **18** | ±0 | ±0 | -3 | -1 | ±0 |
| nomystery11 | **20** | ±0 | ±0 | ±0 | ±0 | ±0 | **14** | ±0 | ±0 | -2 | ±0 | ±0 | **10** | ±0 | -1 | -2 | ±0 | ±0 | 8 | ±0 | ±0 | ±0 | **+3** | +1 |
| openstacks08 | **30** | ±0 | ±0 | ±0 | ±0 | ±0 | **25** | ±0 | ±0 | ±0 | ±0 | ±0 | **24** | ±0 | ±0 | ±0 | ±0 | ±0 | **22** | ±0 | -3 | -2 | ±0 | ±0 |
| openstacks11 | **20** | ±0 | ±0 | ±0 | ±0 | ±0 | **18** | ±0 | ±0 | ±0 | ±0 | ±0 | **17** | ±0 | ±0 | ±0 | ±0 | ±0 | **17** | ±0 | -3 | -3 | ±0 | ±0 |
| openstacks14 | **20** | -1 | -1 | -1 | -1 | -1 | **15** | -2 | -4 | -4 | -2 | -2 | **7** | ±0 | -3 | -3 | ±0 | ±0 | **3** | ±0 | -1 | ±0 | ±0 | ±0 |
| openstacks | **9** | ±0 | -2 | -2 | -2 | -2 | **7** | ±0 | ±0 | ±0 | ±0 | ±0 | **7** | ±0 | ±0 | ±0 | ±0 | ±0 | **7** | ±0 | ±0 | ±0 | ±0 | ±0 |
| parcprinter08 | 17 | -2 | **+1** | -2 | -3 | -3 | 13 | ±0 | **+1** | ±0 | ±0 | -1 | 11 | ±0 | **+2** | ±0 | ±0 | -1 | 11 | -1 | **+2** | **+2** | +1 | ±0 |
| parcprinter11 | 13 | -1 | **+1** | -2 | -2 | -2 | 9 | ±0 | **+1** | ±0 | ±0 | ±0 | 7 | ±0 | **+2** | ±0 | +1 | ±0 | 6 | ±0 | **+3** | +2 | +2 | +2 |
| parking11 | **11** | -1 | -1 | -2 | -3 | -1 | **1** | ±0 | ±0 | ±0 | ±0 | ±0 | **0** | ±0 | ±0 | ±0 | ±0 | ±0 | **0** | ±0 | ±0 | ±0 | ±0 | ±0 |
| parking14 | **14** | -2 | -3 | -6 | -3 | -3 | **4** | ±0 | -3 | -4 | ±0 | ±0 | **0** | ±0 | ±0 | ±0 | **+1** | ±0 | **0** | ±0 | ±0 | ±0 | ±0 | ±0 |
| pathways-nn | **5** | ±0 | ±0 | ±0 | ±0 | ±0 | 4 | ±0 | **+1** | ±0 | **+1** | ±0 | **4** | ±0 | ±0 | ±0 | ±0 | ±0 | 4 | ±0 | ±0 | ±0 | ±0 | ±0 |
| pegsol08 | **30** | ±0 | ±0 | ±0 | ±0 | ±0 | **30** | ±0 | ±0 | ±0 | ±0 | ±0 | 29 | -1 | ±0 | -2 | ±0 | ±0 | 27 | ±0 | ±0 | ±0 | ±0 | ±0 |
| pegsol11 | **20** | ±0 | ±0 | ±0 | ±0 | ±0 | **20** | ±0 | ±0 | ±0 | ±0 | ±0 | 19 | -2 | ±0 | -2 | ±0 | ±0 | 17 | ±0 | ±0 | ±0 | ±0 | ±0 |
| pipes-notank | **45** | ±0 | ±0 | -2 | -30 | -27 | **30** | ±0 | -1 | -5 | -14 | -12 | **22** | ±0 | -2 | -6 | -5 | -5 | 15 | ±0 | -1 | -2 | ±0 | -1 |
| pipes-tank | **35** | -2 | -6 | -11 | -9 | -9 | **20** | ±0 | -3 | -5 | -3 | -3 | **16** | -1 | -4 | -5 | -1 | -1 | 11 | ±0 | -1 | -3 | ±0 | ±0 |
| psr-small | **50** | ±0 | ±0 | ±0 | ±0 | ±0 | **50** | ±0 | ±0 | ±0 | ±0 | ±0 | 49 | ±0 | ±0 | ±0 | **+1** | ±0 | 49 | ±0 | ±0 | ±0 | ±0 | ±0 |
| rovers | 15 | ±0 | **+1** | -2 | -1 | ±0 | 8 | ±0 | **+1** | ±0 | **+1** | ±0 | 6 | ±0 | ±0 | ±0 | **+1** | ±0 | 5 | **+1** | **+1** | **+1** | **+1** | **+1** |
| satellite | 9 | ±0 | **+2** | ±0 | **+2** | ±0 | 7 | ±0 | ±0 | ±0 | **+1** | ±0 | 6 | ±0 | ±0 | ±0 | **+1** | ±0 | 5 | ±0 | ±0 | -1 | **+1** | ±0 |
| scanalyzer08 | 13 | **+1** | ±0 | ±0 | -1 | -1 | **12** | ±0 | ±0 | -3 | ±0 | ±0 | **12** | ±0 | -3 | -3 | ±0 | ±0 | **12** | ±0 | -3 | -3 | ±0 | ±0 |
| scanalyzer11 | **10** | ±0 | ±0 | ±0 | -1 | -1 | **9** | ±0 | ±0 | -3 | ±0 | ±0 | **9** | ±0 | -3 | -3 | ±0 | ±0 | **9** | ±0 | -4 | -3 | ±0 | ±0 |
| sokoban08 | **30** | ±0 | ±0 | ±0 | ±0 | ±0 | 29 | ±0 | ±0 | -1 | ±0 | ±0 | 24 | ±0 | **+3** | ±0 | ±0 | ±0 | 22 | ±0 | **+3** | +1 | ±0 | ±0 |
| sokoban11 | **20** | ±0 | ±0 | ±0 | ±0 | ±0 | **20** | ±0 | ±0 | ±0 | ±0 | ±0 | **20** | ±0 | ±0 | ±0 | ±0 | ±0 | 19 | ±0 | **+1** | -1 | ±0 | ±0 |
| storage | **20** | ±0 | ±0 | -1 | -1 | ±0 | **17** | ±0 | ±0 | -1 | -1 | ±0 | **15** | ±0 | ±0 | ±0 | ±0 | ±0 | 14 | ±0 | ±0 | ±0 | ±0 | ±0 |
| tetris14 | **17** | ±0 | ±0 | ±0 | -15 | -15 | **14** | ±0 | -3 | -4 | -13 | -12 | **11** | -1 | -3 | -3 | -11 | -9 | 9 | ±0 | -4 | -4 | -8 | -7 |
| tidybot11 | **20** | ±0 | ±0 | ±0 | -19 | -19 | **20** | ±0 | -1 | -3 | -19 | -19 | 18 | -1 | -4 | -6 | -17 | -17 | 13 | ±0 | -6 | -8 | -13 | -12 |
| tidybot14 | **20** | ±0 | ±0 | ±0 | -20 | -20 | 18 | ±0 | -2 | -5 | -18 | -18 | 14 | -1 | -6 | -10 | -14 | -14 | 6 | ±0 | -6 | -6 | -6 | -6 |
| tpp | **9** | ±0 | ±0 | ±0 | -1 | ±0 | **7** | ±0 | ±0 | ±0 | -1 | ±0 | **6** | ±0 | ±0 | ±0 | ±0 | ±0 | **6** | ±0 | ±0 | ±0 | ±0 | ±0 |
| transport08 | 17 | ±0 | **+1** | -2 | -1 | ±0 | 15 | ±0 | ±0 | -1 | -2 | ±0 | 12 | **+1** | **+1** | -1 | **+1** | **+1** | 11 | ±0 | ±0 | ±0 | -1 | ±0 |
| transport11 | 15 | ±0 | **+1** | -1 | -2 | -1 | 11 | ±0 | ±0 | ±0 | -2 | -1 | 8 | **+1** | **+1** | -2 | **+1** | **+1** | 6 | ±0 | ±0 | ±0 | **+1** | ±0 |
| transport14 | 13 | **+1** | ±0 | -1 | ±0 | ±0 | **9** | ±0 | ±0 | ±0 | -3 | ±0 | **9** | ±0 | ±0 | -3 | -2 | ±0 | 7 | ±0 | -1 | -2 | -1 | ±0 |
| trucks | **13** | -1 | ±0 | -1 | -1 | -1 | **8** | ±0 | ±0 | ±0 | ±0 | ±0 | **6** | ±0 | ±0 | ±0 | ±0 | ±0 | 5 | ±0 | ±0 | **+1** | ±0 | ±0 |
| visitall11 | 16 | ±0 | **+1** | -1 | ±0 | ±0 | 12 | -1 | ±0 | -1 | ±0 | ±0 | 9 | ±0 | ±0 | ±0 | ±0 | ±0 | 9 | ±0 | ±0 | ±0 | ±0 | ±0 |
| visitall14 | **10** | ±0 | ±0 | -1 | -1 | ±0 | **6** | ±0 | ±0 | ±0 | ±0 | ±0 | **4** | ±0 | ±0 | ±0 | ±0 | ±0 | 3 | ±0 | ±0 | ±0 | **+1** | ±0 |
| woodwork08 | 25 | ±0 | ±0 | -3 | -6 | -11 | 15 | ±0 | -1 | -3 | -7 | -4 | 10 | ±0 | **+1** | -1 | ±0 | -1 | 7 | ±0 | **+2** | **+2** | **+2** | ±0 |
| woodwork11 | **18** | ±0 | -1 | -2 | -3 | -5 | **10** | ±0 | -1 | -3 | -4 | -4 | 5 | ±0 | **+1** | -1 | ±0 | -2 | 2 | ±0 | +2 | +2 | **+3** | -1 |
| zenotravel | **13** | ±0 | ±0 | ±0 | ±0 | ±0 | 10 | ±0 | ±0 | ±0 | **+2** | ±0 | 8 | ±0 | +1 | ±0 | **+2** | ±0 | **8** | ±0 | ±0 | ±0 | ±0 | ±0 |
| Sum all | **1190** | -8 | -20 | -92 | -139 | -143 | **897** | -5 | -16 | -85 | -82 | -84 | **748** | -5 | -22 | -66 | -22 | -53 | 651 | +7 | -20 | -39 | **+13** | -26 |

Table 1: The coverage results as diff from the baseline BnB, for four domain suites defined by the 25%, 50%, 75%, and 100% of best known solution cost for the classical planning task as an OSP task cost bound. bl stands for blind, max$^b$ and max for $h^{max}$, bound-sensitive and regular variants, ms$^b$ and ms for *merge-and-shrink*, bound-sensitive and regular variants, respectively.

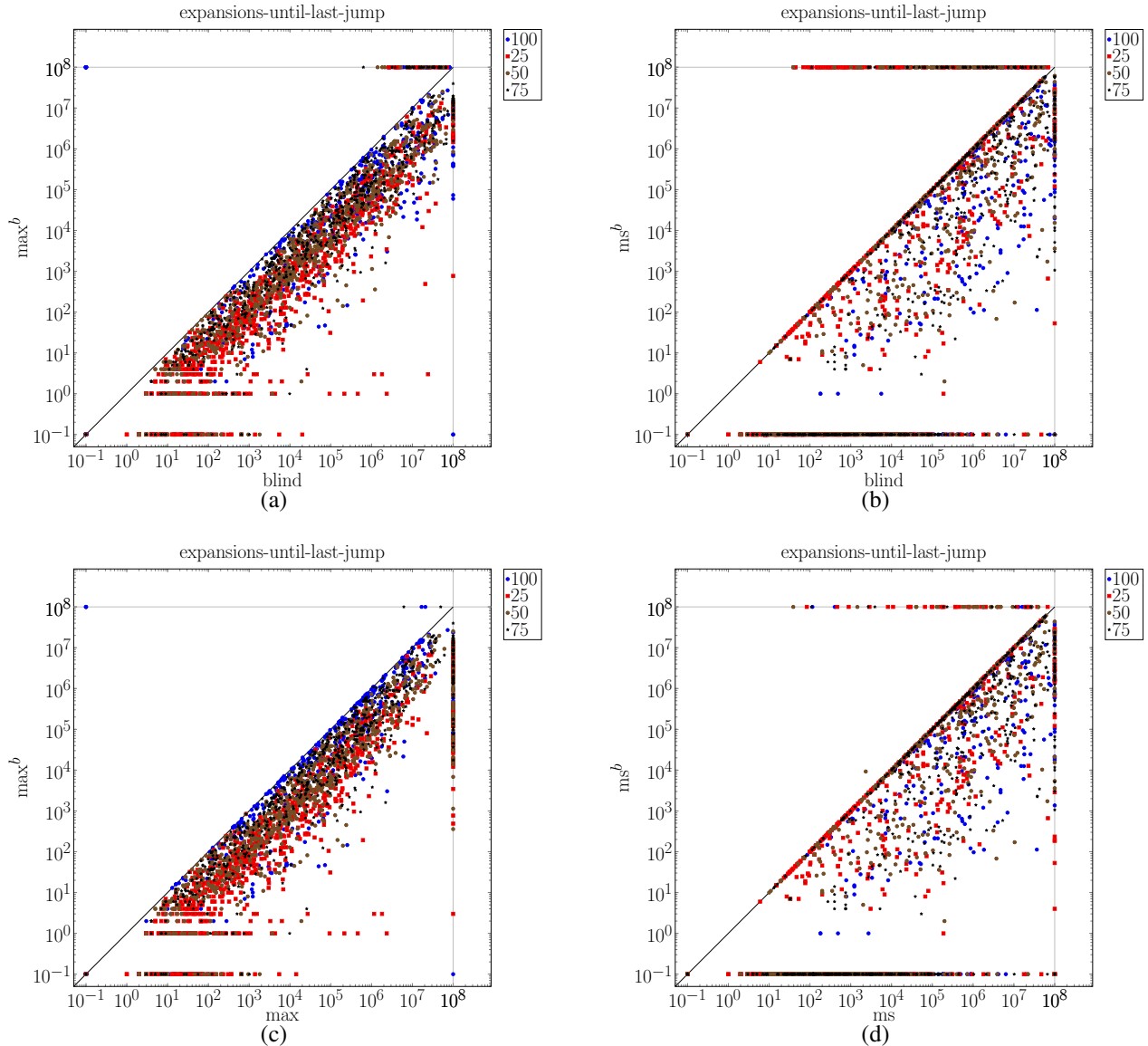

Figure 2: Expansions up to the last layer, $A^*$ with blind heuristic vs. (a) bound-sensitive $h^{max}$ and (b) bound-sensitive *merge-and-shrink*; $A^*$ with bound-sensitive vs. regular heuristic for (c) $h^{max}$ and (c) *merge-and-shrink*.

available OSP benchmarks [Katz *et al.*, 2019b]. The set of benchmarks is taken from the International Planning Competitions of recent years, in which goal facts are replaced with utilities, and the bound set at 25%, 50%, 75%, or 100% of the cost of the optimal or best known solution to each problem. The baseline for our comparison is a blind branch-and-bound search, currently the best available configuration for oversubscription planning that we know of [Katz *et al.*, 2019a]. We compare this baseline to our proposed approach of $A^*$ search on the MCF compilation of the OSP task. Since the compilation introduces intermediate states at which some but not all of the $o^{v,\vartheta}$ have been applied, we use a further optimization that avoids generating these nodes and applies all of the $o^{v,\vartheta}$ actions in a single step, reducing the state space to that of the original OSP task. We experiment with blind $A^*$ search, and

$A^*$ using classical $h^{max}$ and $h^{MS}$, as well as the two heuristics' bound-sensitive variants introduced here. For $h^{MS}$, we used exact bisimulation with an abstract state space threshold of 50000 states and exact generalized label reduction [Sievers *et al.*, 2014]. The experiments were performed on Intel(R) Xeon(R) CPU E7-8837 @2.67GHz machines, with time and memory limits of 30min and 3.5GB, respectively. Per-domain and overall coverage, as well as per-task node expansions for the various configurations and problem suites are shown in Table 1 and Figure 2, respectively. We now report some observations from our results.

- Blind branch-and-bound search usually slightly outperforms blind $A^*$ in terms of coverage, except for the 100% suite. The difference between the two may come

down to the fact that $A^*$ must do extra work in ordering the priority queue, while the variant of branch and bound search that we consider uses no ordering heuristic and can use a simple stack as its search queue. Alternately it may be due to small differences in implementation.

- Bound-sensitive heuristics are much more informative than their classical variants on OSP problems, sometimes decreasing expansions by orders of magnitude. Compared to non-bound-sensitive heuristics, they also almost always result in better coverage.

- Blind search dominates informed search in terms of coverage when bounds are low, but the effect diminishes as the bound increases and it becomes intractable to explore the full state space under the bound. For the 25% suite of problems, heuristic configurations solve an average of approximately 100 instances fewer than the baseline, compared to approximately 15 instances fewer on the 100% suite. Notably, bound-sensitive $h^{\text{MS}}$ has the best coverage in the 100% suite, solving 13 problems more than the baseline, and 6 more than blind $A^*$.

- Coverage on several domains benefits from more informed search schemes. On BLOCKSWORLD, DRIVER-LOG, and MICONIC, bound-sensitive $h^{\text{MS}}$ solves the largest number of problems, and this is also the case for bound-sensitive $h^{max}$ on FLOORTILE, PARC-PRINTER, and SOKOBAN.

- $h^{\text{MS}}$ often times out in the construction phase and before search has begun. This occurs on average in approximately 300 problems per suite, or 1200 problems total. This is especially pronounced in the TIDYBOT, TETRIS, and PIPESWORLD-NOTANKAGE domains. This suggests a hybrid approach that combines the strengths of blind search and $h^{\text{MS}}$: setting an upper bound on the time allotted to heuristic construction, and running blind search instead if construction does not terminate within this bound. Using this configuration with a value of 10 minutes for the upper bound results in a planner that outperforms blind $A^*$ by +11, +16, +37, and +38 instances for the 25%, 50%, 75%, and 100% suites, respectively. This makes $h^{\text{MS}}$ schemes that are less expensive to construct but maintain informativeness in this setting an appealing future subject of research.

## Conclusions and Future Work

We have shown that a previously introduced compilation to multiple cost function classical planning allows the $A^*$ algorithm to be used to solve oversubscription planning problems, and introduced a family of bound-sensitive heuristics that are much more informed than their classical counterparts in this setting. Our experiments show that this approach results in a state-of-the-art method for some bound settings and domains.

One future research direction we would like to explore that builds on the methods introduced here is the use of non-admissible heuristics for satisficing OSP. The method by which bound-sensitive $h^{max}$ is obtained is fairly general and should be equally applicable for $h^{add}$ or general relaxed plan heuristics [Keyder and Geffner, 2008]. A second direction is

the use of these heuristics in other planning settings in which tradeoffs must be made between different cost functions, e.g. minimizing fuel use in the presence of bounds on time or vice versa in logistics problems.

Finally, our methods may be applicable to numeric planning problems in which the variables describe resources that are strictly decreasing and can be expressed in terms of secondary cost functions and associated bounds. Bound-sensitive heuristics could provide a principled way of reasoning about numeric variables in this context.

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
