# OpenReview forum: "A∗ Search and Bound-Sensitive Heuristics for Oversubscription Planning"
_icaps-conference.org/ICAPS/2019/Workshop/HSDIP_

### Official Review · AnonReviewer1 · 2019-03-29
**missing related work - weak accept**

**Rating:** 6
**Confidence:** 4

**Review:**

The paper proposes modifications to admissible heuristics to make them better informed in a multi-criteria setting where one cost function is the minimization objective and one or more secondary cost functions are constrained by bounds. The modified heuristics are applied to a reformulation of oversubscription planning.

Over all it is not a bad paper, but I think it misses some relevant connections. I also have some questions about the OSP formulation.

The special case of multi-criteria planning optimizing for one cost function while remaining within bounds for another has been studied in the context of stochastic problems. The ICAPS 2016 paper by Trevizan et al. introduced algorithms for the constrained SSP (CSSP) problem, which is a stochastic shortest path problem with exactly this kind of constraint/cost structure. More to the point, the ICAPS 2017 paper by Trevizan et al. introduced a form of projection/operator counting heuristics to SSPs, which they also extended to CSSPs. The extension follows essentially the same pattern as that used by the authors of this paper, in that the bounding constraint on each of the secondary costs is added to the heuristic formulation. Clearly this can be applied to non-stochastic problems as well, in which case it reduces to an operator counting heuristic for the bounded multi-criteria problem.

Another special case that has seen some attention is the bounded-cost planning problem. This is formulated the same way as the bounded MCF in this paper but without a primary cost function. In other words, the question is simply does there exist any plan within the secondary cost bound? Typically, this problem considers only a single bounding cost function. Some specialized search algorithms were introduced by Stern et al. (ICAPS 2011) and Thayer et al. (ICAPS 2012), but adaptation of some common planning heuristics to this setting were also proposed (Haslum ICAPS 2013; Dobson and Haslum HSDIP 2017). Again, the pattern of adaptation is similar, with the cost bound somehow imposed on the selection of actions in the abstract or relaxed plan.

The paper should at least discuss these closely related works. Even better would be a comparison between the proposed new heuristics and the previous ones in settings where they are comparable. For example, the bounded-cost problem can be formulated as an OSP, by simply making the goal soft, which has a solution with reward equal to the trivial upper bound (the sum of all subgoal utilities) if and only if the original bounded-cost problem is solvable.

The OSP formalism used in this paper, and presumably also in the paper by Katz et al. cited for the reformulation, assigns utilities only to individual facts, i.e., variable-value equalities. There is no explicit provision for assigning utilities to conjunctions (or disjunctions) of facts (for example, to say that the utility of have(bread) and have(butter) is more than the sum of the utilities of each of the two facts by themselves, or, for that matter, that the utility of have(train-ticket) and have(bus-ticket) is no more than the max of the utilities of each of these two facts individually). One can imagine
encodings that use an artificial, zero-cost action to set an auxiliary variable to true when a conjunction is achieved, but this raises some problems with the reformulation, in that undoing any part of the conjunction must also force a reset of the auxiliary variable. It is also not clear how this would work in situations where the utility of a conjunction is less than the sum of its parts. It would be good if the authors can comment in the paper on how limiting the restriction to single-fact utilities is.

The readability of Table 1 could be enhanced. For example, alternating rows with white and lightly shaded backgrounds would make it visually easier to follow a row. The plus/minus zero entries could be omitted (blank) to make it easier to identify where the differences are.

References:

Felipe Trevizan, Sylvie Thiébaux, Pedro Henrique Santana, Brian Charles Williams. Heuristic Search in Dual Space for Constrained Stochastic Shortest Path Problems. ICAPS 2016. http://www.aaai.org/ocs/index.php/ICAPS/ICAPS16/paper/view/13179

Felipe W. Trevizan, Sylvie Thiébaux, Patrik Haslum. Occupation Measure Heuristics for Probabilistic Planning. ICAPS 2017. https://aaai.org/ocs/index.php/ICAPS/ICAPS17/paper/view/15771

Jordan Tyler Thayer, Roni Stern, Ariel Felner, Wheeler Ruml. Faster Bounded-Cost Search Using Inadmissible Estimates. ICAPS 2012. http://www.aaai.org/ocs/index.php/ICAPS/ICAPS12/paper/view/4706

Roni Tzvi Stern, Rami Puzis, Ariel Felner. Potential Search: A Bounded-Cost Search Algorithm. ICAPS 2011. http://aaai.org/ocs/index.php/ICAPS/ICAPS11/paper/view/2687

Patrik Haslum. Heuristics for Bounded-Cost Search. ICAPS 2013. http://www.aaai.org/ocs/index.php/ICAPS/ICAPS13/paper/view/5993

Sean Dobson, Patrik Haslum. Cost-Length Tradeoff Heuristics for Bounded-Cost Search. HSDIP 2017. http://icaps17.icaps-conference.org/workshops/HSDIP/proceedings/dobson-haslum-icaps2017wshsdip.pdf

I think I have seen a paper titled something along the lines of "planning with conjunctive utilities" somewhere, but now I cannot find it or recall where, or who wrote it.

---

> ### Author Response · Authors · 2019-04-11
> **We thank the reviewers for their constructive feedback.**
>
> The restriction to single fact utilities here is not an essential limitation. As in the Keyder & Geffner compilation that the approach here is based on, conjunctive utilities can be handled with auxiliary operators that achieve a hard goal associated with each utility formula. Some changes then need to be made to the rest of the problem: 1) a new precondition is introduced for the original problem actions 2) an END action is introduced that deletes this precondition and adds a precondition required by the newly introduced auxiliary actions. This forces the plan to be partitioned into 2 phases separated by the END action, the first of which consists of original plan actions and the second of which consists of the auxiliary actions, and prevents changes to the original problem variables once the END action has been applied. In cases where the utility of a conjunction is less than the sum of its parts, the parts would also have to be stated as exhaustive formulae (e.g. util(A and not B) = 10, util(not A and B) = 10, util (A and B) = 5). The utility of a state, as before, would be given by the summed utility of the satisfied formulae.
>
> We can briefly state that our approach can be extended to handle more complicated utilities, but will not go into much more detail due to 1) lack of space 2) these issues being dealt with more fully in the Keyder & Geffner paper, and being only peripherally related to the central ideas here.

---

> > ### Comment · AnonReviewer1 · 2019-04-11
> > **Page limit is 9 pages**
> >
> > "lack of space" is a poor excuse. The workshop cfp says to keep papers to at most 9 pages including references. That's another 2 pages and a bit that could be added to this paper.
> >
> > That said, I also don't think that the handling of conjunctive goal utilities needs more than a brief comment. It is, as the authors also say, well known how they can be compiled into action costs. What is missing from the paper is the observation that the 2-phase encoding trick (which is described as an efficiency hack in the paper) becomes a necessity when goal satisfaction is not direct.
> >
> > The extra 2 pages should be used to discuss the missing related work and compared the proposed heuristics to those already in the literature for the same and very similar problems.

---

### Official Review · AnonReviewer2 · 2019-04-06
**somewhat incremental; clarity could be improved; topic fits workshop**

**Rating:** 6
**Confidence:** 3

**Review:**

The paper presents an approach to solve oversubscription planning
(OSP) tasks optimally by using a translation to classical planning
with multiple cost functions. A new class of heuristics is introduced
that makes explicit use of the cost bounds encoded during the
transformation such that solving the tasks optimally with the standard
A* + admissible heuristic approach also leads to an optimal solution
of the OSP task.

The main contribution of the paper is the introduction the new type of
heuristic based on the already existing task transformation, and
designing the necessary adaption for two concrete heuristics, namely
hMax and Merge&Shrink heuristics.


Some comments:
In my opinion, the authors could do a better job introducing the task
reformulation, since this is the basis for the rest of the paper. I
found it somewhat hard to understand why bounds are implemented in
this way.

The introduction of Merge&Shrink lacks formality, leading to the fact
that Theorem 3, i.e., the claim that the introduced adaption of
standard M&S heuristics is indeed an admissible heuristic that
respects the bounds of the transformed task properly, cannot be
verified. There are too many details missing.

The experimental evaluation is not very convincing. Although the
search space size gets reduced significantly, coverage is
significantly worse than for the baselines branch-and-bound, as well
as blind search. There is only a single setting where the new M&S
heuristic is better. Other than that, there is only a few domains
where the new heuristics can solve more instances.

You state that M&S times out when constructing the heuristics in many
cases. 1) Does this only happen in instances that are too large to be
solved, anyway? 2) Given the large amount of existing M&S
configurations, what is the bottleneck of trying out a different
strategy? Are there any specifics that need to be adapted for each
variant individually?


Minor:
- please remove the copyright statements from the first page
- "in which a set of hard goals" => "where a set of hard goals"
- "shown to *be* outperform"
- I found the usage of "v = v_g" of values as v_g a bit confusing, since
in the rest of the paper you use \vartheta. Also, later you use \theta
instead. Please make it consistent throughout the paper.
- in the definition of G' of Def1: it should probably be "v = v_g", not v_g
- Please consider reformulating the sentence ".. when the node n with
minimal f(n) value is a goal state".
- "if the cost of the new path is less" => lower
- M&s section: "s, s' *C_i*
- you never introduce the configuration shortcuts "ms^b", "max^b", and
so on.
- "80 instances *fewer* than" => less than

---

> ### Author Response · Authors · 2019-04-11
> **We thank the reviewer for their comments.**
>
> The main contribution of this work is to show that OSP problems can be solved with A*. To the best of our knowledge, this has not been not done before, with previous approaches employing the Branch&Bound search algorithm. One of the benefits of switching to A* is the ease of adaptation of existing techniques from classical planning, such as search space pruning techniques.
>
> We’re happy to restructure the discussion of the compilation, though would appreciate more detail as to where the reviewer thought it was lacking and required more intuition. Most planning approaches natively handle action costs, so it seems natural to us to try to restate the OSP problem in those terms. It was not obvious previously that standard planning algorithms could be adapted to this setting, and restating the problem in these terms allows at least a subset of standard planning approaches to be applied to OSP.
>
> The focus of this paper is not M&S, and we therefore did not include too much detail to stay within page limits. The basic idea behind the soundness of the approach here is common to all abstractions and not specific to M&S: for any path through the state space of the original problem, there is a path in the abstract state space of equal or lower cost, and this property is sufficient to prove both admissibility of the primary cost estimate and soundness of pruning based on the secondary costs in the abstract space.
>
> The instances on which M&S construction times out are not necessarily instances where search would time out anyway. There are problems that can be solved with very few expansions (e.g. < 1000) by blind search that are not solved by M&S since construction times out. A trivial approach to improve upon our current results would be then to try M&S construction until some time limit and then either use the abstraction created so far or run blind search if construction has not terminated. These results could be included, we have not shown such a configuration in the current version of the paper as we were focusing on the high-level idea of solving OSP with classical planning methods. We have experimented with some other M&S strategies, but have not yet found a lightweight one that works well in this setting.

---

> > ### Comment · AnonReviewer2 · 2019-04-11
> > **Thank you for the response**
> >
> > It would probably help if you discussed the example of Figure 1 a bit earlier, when introducing the reformulation. Additionally, it would help to also show the reformulated state space, with the new "buy-utility" actions.
> >
> > Again, lack of space is not a concern for the workshop, so you could introduce M&S more formally. I got the intuition, but you should probably call this a "proof sketch", not a proof, if you don't give more details.

---

### Meta-Review · Program_Chairs · 2019-04-25

**Recommendation:** Accept
**Confidence:** 5

**Metareview:**

Dear Authors,
thank you very much for your submission. We are happy to inform you that
we have decided to accept it and we look forward to your talk in the workshop.
Please, go over the feedback in the reviews and correct or update your papers
in time for the camera ready date (May 24). In particular, please address the
comments raised by Reviewer1 regarding clarity and discussion of related work,
by making use of the additional space (9 pages) allowed by HSDIP.
Best regards
HSDIP organizers